# Depiction of the Genetic Alterations and Molecular Landscapes of Thymic Epithelial Tumors: A Systematic Review and Meta-Analysis

**DOI:** 10.3390/cancers16172966

**Published:** 2024-08-25

**Authors:** Xin Wang, Hongming Jin, Xiaotong Feng, Zhijian Liang, Ruoyi Jin, Xiao Li

**Affiliations:** 1Thoracic Oncology Institute, Peking University People’s Hospital, Beijing 100044, China; wangxin0629@stu.pku.edu.cn (X.W.); 1810301223@pku.edu.cn (H.J.); 2010301207@stu.pku.edu.cn (X.F.); liangzhijianlzj@pku.edu.cn (Z.L.); 2110305326@bjmu.edu.cn (R.J.); 2Department of Thoracic Surgery, Peking University People’s Hospital, Beijing 100044, China

**Keywords:** thymic epithelial tumors, gene mutation, PD-L1, meta-analysis, systematic review

## Abstract

**Simple Summary:**

Previous studies have reported the gene mutational landscapes and PD-L1 expression levels of thymic epithelial tumors. However, due to the rarity of the disease, heterogeneity, whether from the different sequencing methods or the patients, does exist across these studies, hampering the large-scale application of targeted therapy and immunotherapy. Thus, we conducted a meta-analysis on the genetic alterations and molecular landscapes of TETs and obtained the pooled estimated mutation rates of the most frequently mutated genes in TETs, as well as the PD-L1 expression levels in TETs. Confounding factors that may contribute to the heterogeneity are also discussed in our manuscript. Our efforts have provided a more accurate overview of the gene mutation landscape and PD-L1 expression levels in TETs, which may contribute to the cure of TETs, advanced TETs, or metastatic TETs in particular.

**Abstract:**

Thymic epithelial tumors (TETs), consisting of thymomas, thymic carcinomas (TCs), and thymic neuroendocrine tumors, are rare diseases. Surgery remains the prime option in resectable and early-stage TETs, while chemotherapy, targeted therapy, and immunotherapy are also potential treatment modalities. However, the inadequate comprehension of the molecular landscape of TETs impedes the exploitation of such therapies. Hence, we conducted a meta-analysis which includes 21 studies reporting on genomic alterations in TETs and 14 studies reporting on PD-L1 expression levels, respectively. The pooled estimated rates of the most frequently mutated genes and PD-L1 expression levels were analyzed using the R software. We uncovered that the pooled estimated overall mutation rate is 0.65 ([0.49; 0.81]), and the top three genes with highest mutation frequency in thymomas and TCs are *GTF2I* (0.4263 [0.3590; 0.4936]), *TP53* (0.1101 [0.0000; 0.2586]), and *RAS* (0.0341 [0.0104; 0.0710]), and *TP53* (0.1797 [0.0732; 0.3203]), *CDKN2A* (0.0608 [0.0139; 0.1378]), and *TET2* (0.0318 [0.0087; 0.0639]), respectively. A uniform *GTF2I* mutational rate in thymomas and *TP53* mutational rate in thymic squamous cell carcinomas (TSCCs) are also observed. The pooled estimated expression level of PD-L1 is 0.71 ([0.59–0.81]). This systematic review provides an overview of the gene alteration landscape and PD-L1 expression levels in TETs, discovers several potential confounding factors that may contribute to the high heterogeneity, and facilitates deeper investigations into the elucidation of the molecular landscape of TETs.

## 1. Introduction

Thymic epithelial tumors (TETs), including thymomas, thymic carcinomas (TCs), and thymic neuroendocrine tumors (NETs), stand as the predominant tumors in the anterior mediastinum [1], while the incidence rate is relatively low compared to other common thoracic tumors [2]. Thymomas have been classified into the following five histological subtypes by the WHO: type A, AB, B1, B2, and B3 [3]. Higher histological subtypes correlate with more advanced stages, while TCs tend to metastasize [4]. Surgery remains the standard therapy for early-stage resectable TETs, while platinum-based chemotherapy is the standard therapy for unresectable and advanced TETs. However, alternative strategies, including targeted therapy and immunotherapy for non-responders to chemotherapy, are still challenging [5]. Scientists have achieved much progress in promoting the efficiency of targeted therapy and immunotherapy in TET patients. Clinical trials aiming at testing the efficiency of targeted drugs [6,7] and anti-PD-1/PD-L1 drugs [8] have been initiated. But due to the rarity and heterogeneity of this disease, these therapies for advanced and refractory TETs are hard to implement on a larger scale, and the response rate is relatively low.

Thanks to the advances of next-generation sequencing (NGS), including whole-exome sequencing (WES), whole-genome sequencing (WGS), targeted NGS, etc., scientists have made tremendous efforts to uncover the genetic characteristics of TETs [9]. One of the most astonishing breakthroughs is the study by Radovich et al., reporting the mutational rates of *GTF2I*, *HRAS*, *NRAS*, and *TP53*. While the mutation of *GTF2I* was mainly found in type A (100%) and AB thymomas (70%), *TP53* mutation was mainly spotted in type B thymomas and TCs in correlation with poorer survival [10]. Later studies reporting the gene mutational landscapes of TETs varied significantly in the mutated genes and their mutation proportions.

Effective immune response could eradicate malignant cells. However, cancer cells have evolved multiple mechanisms to evade immune surveillance, resulting in the impaired effector function of immune cells and the loss of anti-tumor immune response. Immunotherapy has emerged as a breakthrough in the field of cancer treatment, aimed at enhancing natural defenses to eliminate malignant cells. A variety of solid tumors have shown good clinical responses to immunotherapy, such as melanoma, renal cell carcinoma, breast cancer, and non-small-cell lung cancer [11].

Programmed cell death protein 1 (PD-1) and its ligand (PD-L1) are immune checkpoint receptors and play a major role in regulating effector T cell activity in tissues [12]. A meta-analysis of neoadjuvant immunotherapy for early non-small-cell lung cancer found that a high expression of PD-L1 was associated with higher MPR and pCR levels in immunotherapy [13]. Some studies have shown variable PD-1 and PD-L1 expression levels, suggesting the potential for immune checkpoint inhibitor (ICI)-based immunotherapy in TETs [14,15], though the prognostic value of PD-1/PD-L1 expression levels remains unclear.

As mentioned above, the genomic alternations and the expression levels of PD-L1 show great discrepancy in different studies, and the results might impact the clinical implementation of corresponding therapies. Therefore, we performed a systemic review and meta-analysis to explore the dynamic genomic alternations and the expression levels of PD-L1 in TET patients of different subtypes, while spotting potential confounding factors that might lead to variable results. This review can contribute to a better understanding of the molecular alternations in TET patients and support further clinical practice when considering the application of targeted therapy and ICIs.

## 2. Materials and Methods

### 2.1. Search Strategy

We identified studies reporting gene alterations and expression levels of PD-L1 in TET patients by performing a thorough search in four databases (PubMed, Embase, Web of Science, and Cochrane Library) for papers published between 1 January 2000 and 31 October 2023. The following three sets of keywords were used: #1: “Thymus Neoplasms” OR “Thymic Epithelial Tumor” OR “Thymoma” OR “Thymic Carcinoma” OR “Thymic Neuroendocrine Tumor” OR “Thymic Cancer”; #2: “Gene” OR “Cistron” OR “Cistrons” OR “Genetic Materials” OR “Genetic Material” OR “genetic feature” OR “genetic characteristics” OR “genetic characteristic” OR “genetic features” OR “Genomic alteration” OR “Genomic alterations”; #3: “PD-L1” OR “programmed cell death 1 ligand 1 protein” OR “PDL1” OR “CD274” OR “B7-H1” OR “B7H1” OR “PD-1” OR “Programmed cell death protein 1” OR “PD1” OR “CD279”; “#1 AND #2” and “#1 AND #3” were searched separately.

### 2.2. Selection Criteria

Two authors (X.W. and H.J.) independently screened all of the retrieved papers, excluded the papers that did not meet any of our criteria, and assessed the rest of the papers to distinguish the ones that fit our criteria based on the Preferred Reporting Items for Systematic Reviews and Meta-Analyses (PRISMA) guidelines [16] (see Appendix A for the checklist). Our inclusion criteria included the following: (1) original studies that report gene mutations and/or PD-L1 expression levels in TETs; (2) samples obtained from surgery or tissue biopsy; (3) studies that adopt targeted NGS or WES to detect potential gene mutation characteristics in TETs (for the gene mutation cohort), or immunohistochemistry (IHC) to report PD-L1 expression levels (for the PD-L1 cohort). Our exclusion criteria for the gene mutation cohort included the following: (1) studies not written in English; (2) insufficient data for further analysis; (3) studies focusing on genes associated with myasthenia gravis (MG). Our exclusion criteria for the PD-L1 expression level cohort included the following: (1) studies not written in English; (2) insufficient data for further analysis; (3) IHC findings not clarified or not based on the tumor proportion score (TPS); (4) a cut-off value that was not 1% or 50%. Studies entering the next stage were discussed by all of the authors in our group for full-text assessment to confirm their eligibility.

### 2.3. Data Extraction

We extracted the following data from the selected papers: authors, nationality of the author(s), year of publication, cohort size, genomic characteristics and pathological details of each patient, methodological details, and relevant statistical findings. Three authors (X.W., H.J., and X.F.) independently gathered these data. Disagreements were resolved through discussion, where all data were collectively reviewed in the presence of all authors. As for the gene mutation cohort, we first carefully examined the reported gene mutations and the tested gene panels in each article, and listed them in a table. Then, the gene mutation frequency was calculated by dividing the number of mutated samples by the total number of samples in which the genes were tested. Each author presented their data, and discrepancies were addressed by re-examining the original studies until a consensus was reached.

### 2.4. Statistical Analysis

We first summarized the rates of gene alterations and PD-L1 expression levels in the selected papers. After data transformation and normality checks using different methods, we pooled the mutation rates of the top six frequently mutated genes and the PD-L1 expression levels using random effect or common effect models. The significance of the pooled estimated rates was determined using the Z test, with a *p*-value < 0.05 considered to be statistically significant. To explore the sources of heterogeneity, we also conducted subgroup analyses and sensitivity analyses. Subgroup analyses were conducted based on factors identified in the previous literature, including the regions, subtypes, and sequencing methods that may influence the gene mutation rates, as well as the cut-off values and types of antibodies which may affect the PD-L1 expression levels. In addition, we defined the PD-L1 expression levels based on the positive proportion of PD-L1 staining in tumor cells (tumor proportion score, TPS). According to three previous clinical trials, we performed the subgroup analysis with 1% and 50% TPS cut-off values [8,17,18]. Egger’s test was used to detect potential publication bias across the studies. All data analyses were performed using the R software “https://www.r-project.org/ (accessed on 1 November 2023)” for our data analysis (version 4.2.3).

### 2.5. Quality of Evidence

The 35 studies included in our systematic review were all cross-sectional studies. The list of the included articles was determined by the authors through discussion and agreement. The way to assess the methodological quality of the included articles was recommended by the Agency for Healthcare Research and Quality (AHRQ) from 11 perspectives with the Cross-sectional/Prevalence Study Quality. This systematic review was registered in the PROSPERO database from the Centre of Reviews and Dissemination, University of York, UK (registration number: CRD42024566519).

## 3. Results

### 3.1. Literature Selection

A total of 4201 articles were first included in our meta-analysis after reviewing 5740 studies reporting the gene alterations in thymic epithelial tumors ranging from 1 January 2000 to 31 October 2023, and removing 1539 duplicate records from PubMed, Embase, Cochrane Library, and Web of Science databases, and independently screened by two researchers. Next, reviews, case reports, editorials, comments, conference abstracts, and studies that failed to meet any of our criteria were removed. Forty studies entered the stage of full-text assessment, during which eight studies were deleted from our cohorts due to the language of the study, incomplete data for further analysis, or the focus of the studies. Finally, 21 studies concentrating on the depiction of the gene mutation landscape in TETs remained for the meta-analysis. For the PD-L1 expression level cohort, 306 studies were initially identified, and 52 duplicates were removed. After similar exclusions, 132 studies underwent full-text assessment, and 14 studies were included for the meta-analysis (Figure 1) [16].

### 3.2. Overall Characteristics

#### 3.2.1. Gene Alteration Cohort

In this cohort, 21 studies concentrating on the gene mutational landscape of TETs investigated a total of 1102 subjects, including 510 thymoma subjects and 592 thymic carcinoma (TC) subjects, and all of the samples were collected from surgeries or biopsies for further detection [10,19,20,21,22,23,24,25,26,27,28,29,30,31,32,33,34,35,36,37,38]. Targeted next-generation sequencing (targeted NGS) was the main method to analyze gene alterations, while only four studies adopted the method of whole-exon sequencing (WES) for detection. Particularly, one study utilized both targeted NGS and WES to cover more comprehensive possible mutational targets and to compare the consequences between these two methods [19]. The regions of the studies varied significantly, including five studies from the US, two studies from Europe, and the remaining fourteen from Asia (Table 1).

#### 3.2.2. PD-L1 Expression Level Cohort

A total of 1214 samples obtained from biopsy or surgery from 14 studies were included in this cohort, and immunohistochemical tests (IHC) were used to examine the expression status of PD-L1 (Table 2) [39,40,41,42,43,44,45,46,47,48,49,50,51,52]. Of note, two studies used multiple antibodies for the IHC tests and analyzed the samples using both 1% and 50% cut-off values for the PD-L1 tumor cell proportion (TPS) [39,45].

### 3.3. Meta-Analysis of Gene Mutational Landscape

#### 3.3.1. Overall Mutation Rates

Among the 21 studies reporting gene alterations in TETs, 18 studies adopted the method of NGS or WES, while 3 studies utilized PCR and direct sequencing to spot mutated genes, which only entailed a limited number of genes, and thus they were excluded when analyzing the overall mutation rates. A total of 633 out of 987 samples (64.13%) were reported to harbor at least one gene alteration. Precisely 320 out of 514 thymoma samples (62.26%) and 313 out of 473 TC samples (66.17%) were found to have gene mutations. The meta-analysis indicates that the pooled overall mutation rate is 0.6546 (95% CI [0.4860; 0.8053], random effect model) (Figure 2A,B). However, high heterogeneity was identified in this cohort (*I*^2^ = 97%, *p* < 0.01). Since these studies were from various regions and adopted different sequencing methods, a subgroup analysis was performed according to the regions and their sequencing methods (Figure 2C,D). In the studies utilizing WES, the mutation rate appeared to be higher (0.87 (95% CI [0.54; 1.00]) than in the targeted NGS (0.59 (95% CI [0.40; 0.76])), but the difference was not statistically significant (*p* = 0.12). Subgroup analysis also showed that the studies from Europe reported a higher mutational rate than the studies from other regions, despite the insignificant difference (*p* = 0.94). To ensure the robustness of the pooled rates, a leave-one-out sensitivity analysis was conducted. The results showed minimal change when any single study was excluded, indicating that no individual study unduly influenced the overall findings (Figure 2E).

As thymomas and TCs show a great discrepancy in terms of their molecular landscapes, biological performances, and clinical characteristics, we next conducted our meta-analysis of gene alterations in thymomas and TCs, respectively (Table 3).

#### 3.3.2. Thymoma

*GTF2I* exhibited the highest mutation frequency in the reported records (86/201, 42.79%), followed by *TP53* (34/435, 7.82%), *RAS* family genes (25/455, 5.49%), *ATM* (9/282, 3.19%), *EGFR* (8/328, 2.44%), and *KIT* (4/264, 1.52%). Next, we performed meta-analyses of the mutation rates of the top six frequently mutated genes in thymomas with different models (Table 3).

*GTF2I* mutation

Five studies document the incidence of *GTF2I* mutation, and the pooled estimated mutation rate is 0.44 [0.34; 0.53] (common effect model). The heterogeneity is relatively low in the *GTF2I* group *(I*^2^ = 38%, *p* = 0.17). Compared to the other genes, the mutation rate is fairly high in thymomas, with a low heterogeneity, which again confirms the high prevalence of the *GTF2I* mutation in thymomas (see Figure 3A for the forest plot and Figure 3B for the funnel plot).

*TP53* mutation

An aggregate of eleven studies reports on the *TP53* mutation rates, and the pooled estimated mutation rate is 0.1101 [0.0000; 0.2586] (random effect model), with extremely high heterogeneity (*I*^2^ = 94%, *p* < 0.0001). Among the eleven studies, only one study utilized WES, and another single study utilized PCR to detect the *TP53* mutation. Additionally, only one study was from Europe. Due to the limited number of subgroups, a subgroup analysis could not performed either by sequencing methods or by regions. The sensitivity analysis reveals that, when omitting the study “Szpechcinski 2022” [35], the *I*^2^ can be reduced to 4.1%, indicating that confounding factors may exist that influence the detection of the *TP53* mutation rate in this study (see Figure 4A for the forest plot, Figure 4B for the funnel plot, and Figure 4C for the sensitivity analysis).

*RAS* family gene mutation

The *RAS* family gene mutation, commonly involving the *HRAS*, *NRAS*, and *KRAS* mutations, is reported in 12 studies, and the pooled estimated mutation rate is 0.0341 [0.0104; 0.0710] (random effect model). Relatively high heterogeneity is spotted in this group (*I*^2^ = 66%, *p* = 0.007) (see Figure 5A for the forest plot and Figure 5B for the funnel plot). The subgroup analysis shows that the mutation rate of the *RAS* family genes in the “WES” subgroup is significantly higher than the “targeted NGS” subgroup (*p* < 0.01), despite the limited number of studies in the “WES” subgroup (only two studies). As for the regions, the mutational rate seems consistent across the three subgroups (see Figure 5C,D for the subgroup analysis). The sensitivity analysis indicates the robustness of the pooled mutation rate of the *RAS* family genes in thymomas (see Figure 5E for the sensitivity analysis).

A meta-analysis was also performed on the prevalence of *ATM*, *EGFR*, and *KIT* mutations in thymomas, and the results are presented in Table 3 (see Figure 6A–C for the forest plots and funnel plots of the *ATM*, *EGFR*, and *KIT* mutation rates in thymomas, respectively). Comparatively low heterogeneity was found in these meta-analyses.

#### 3.3.3. Thymic Carcinoma

Our analysis discovered that the mutation rate of *TP53* was ranked first (129/517, 24.95%) among TCs, while *CDKN2A* was ranked closely behind (79/448 17.63%). The prevalence of the *TET2* mutation was ranked third (19/285, 6.67%), followed by the *RAS* family (35/578, 6.06%), *KIT* (29/482, 6.02%), and *DNMT3A* (6/275, 2.18%). Likewise, further meta-analyses were conducted on the mutational rates of these genes in TCs (Table 3).

*TP53* mutation

The *TP53* mutation rate is comparatively higher in TCs (the pooled estimated rate is 0.1797 [0.0732; 0.3203], random effect model) than in thymomas according to the 16 articles that reported on the TP53 mutation rates, while a high heterogeneity is observed in this group (*I*^2^ = 91%, *p* < 0.0001) (see Figure 7A for the forest plot and Figure 7B for the funnel plot).

To explore the origin of the high heterogeneity, we further classified the samples into several subgroups due to their histologic types, which contain thymic squamous cell carcinoma (TSCC), adenocarcinoma, basaloid carcinoma, neuroendocrine carcinoma, etc. In particular, five studies provided sufficient information on TC subtypes, allowing for further analysis, and the meta-analysis of the TP53 mutation rate was performed on TSCCs. The pooled estimated mutation rate of the *TP53* mutation in TSCCs is 0.1556 [0.0866; 0.2246] (common effect model), which is slightly lower than that in TCs, with a low heterogeneity (*I^2^* = 34%, *p* = 0.1923) (Figure 7C). The data presented in these studies were insufficient for us to perform meta-analyses in other TC histologic subtypes.

Next, we conducted a subgroup analysis of the *TP53* mutation rate in TCs by the sequencing methods they utilized. As shown in Figure 7D, 13 studies adopted targeted NGS to detect the *TP53* mutation, and the pooled estimated rate is 0.1852 [0.0595; 0.3600] (random effect model), with a high heterogeneity (*I^2^* = 92%, *p* < 0.01) in this subgroup. In the WES subgroup, containing four studies, the pooled estimated rate is 0.1600 [0.0603; 0.2956] (common effect model), with a fairly low heterogeneity (*I*^2^ = 0%, *p* = 0.48). The mutation rate was estimated to be higher in the targeted NGS group, but the difference was not statistically significant (*p* = 0.7988, random effect model). The subgroup analyses by regions indicate that the mutational rate of *TP53* in TCs in Europe and North America is much higher than in Asia (*p* = 0.02). The sensitivity analysis does not imply that the exclusion of any specific study would substantially reduce the value of *I*^2^ (Figure 7F).

*CDKN2A* mutation

A total of 16 articles report on the rates of the *CDKN2A* mutation, and the overall effect rate is 0.0608 [0.0139; 0.1378] (random effect model), with an extremely high heterogeneity (*I*^2^ = 91%, *p* < 0.0001) and with similarity to the *TP53* alteration rate in TCs (see Figure 8A for the forest plot and Figure 8B for the funnel plot). As only one study employed WES to detect the *CDKN2A* mutation, conducting further subgroup analysis by this sequencing method was not feasible. No significant difference is found across the three subgroups according to the subgroup analysis by regions (*p* = 0.35) (Figure 8C). Of note, the sensitivity analysis reveals that, when omitting the study “Girard 2022” [22], the *I*^2^ can be reduced to 58% (Figure 8D).

*TET2* mutation

The aggregated mutation rate of *TET2* in TCs is calculated as 0.0318 [0.0087; 0.0639]] (common effect model), consolidating data from eight studies that document the incidence of the TET2 mutation. Minor heterogeneity is observed within this group (*I*^2^ = 37%, *p* = 0.1363) (see Figure 9A for the forest plot and the funnel plot)

A meta-analysis was also performed to assess the prevalence of the *RAS* family, *KIT*, and *DNMT3A* mutations in TCs, with the results summarized in Table 2 (see Figure 9B–D for the forest plots and the funnel plots). Minor heterogeneity is seen across these meta-analyses.

### 3.4. Meta-Analysis of the PD-L1 Expression Level

#### 3.4.1. Overall PD-L1 Expression Status

In 14 eligible studies, the PD-L1 expression statuses were evaluated by the percentage of tumor cells that were PD-L1-expression-positive, which varied from 27% to 89%, while the pooled estimated expression level of PD-L1 in TETs was 71% (95% CI, [0.59; 0.81], *p* < 0.01, random effect model) (Figure 10). Subgroup analyses were performed from the following perspectives (Figure 11A–E).

#### 3.4.2. Subgroup Analysis: Cut-Off Value

Considering the differences in the cut-off value, we divided the 14 studies into two subgroups based on their cut-off values. Eleven studies that adopted “TPS ≥ 1%” (tumor proportion score, TPS) as their cut-off value were classified into the low-threshold group, while the high-threshold group contained ten studies that defined “TPS ≥ 50%” as their cut-off value. Notably, seven studies both adopted 1% and 50% as their cut-off values while analyzing the PD-L1-positive expression ratio [39,41,43,45,49,51,53].

The low-threshold group contains 968 patients from 11 studies, while the high-threshold group contains 962 patients from 10 studies. The pooled expression level of PD-L1 in the low-threshold group (78%) is higher than the high-threshold group (43%), and the difference is significant (*p* < 0.01, random effect model) (Figure 11A).

#### 3.4.3. Subgroup Analysis: Histologic Subtypes

We next performed a meta-analysis of different histologic subtypes, dividing the 14 studies into two subgroups, including the thymoma and the TC groups.

In the thymoma group, including eight studies investigating 675 thymoma patients and using “TPS ≥ 1%” as the threshold, the meta-analysis shows a pooled PD-L1-positive ratio of 83% (95% CI, [0.65; 0.93], *I*^2^ = 95%, random effect model), which is relatively higher than that in the TC group, which includes 207 TC patients from eight studies (71%, 95% CI [0.64; 0.77], random effect model). However, the subgroup difference is not significant (*p* = 0.19) (Figure 11B).

A meta-analysis was also conducted to evaluate the PD-L1 expression level in studies that adopted “TPS ≥ 50%” as the threshold. In seven studies investigating 692 thymoma patients, the pooled PD-L1-positive ratio is 42% (95% CI [0.28; 0.59], random effect model), and the rate in the TC group including 339 patients from ten studies is 41% (95% CI [0.31; 0.52], random effect model) (Figure 11C).

#### 3.4.4. Subgroup Analysis: Antibodies

Several studies indicated that different primary antibodies utilized in the studies would influence the results. To clarify the impact of antibodies in IHC tests on the results, we also conducted meta-analyses on both the low-threshold group and the high-threshold group based on the different choices of antibodies. Five antibodies were used in the 14 eligible studies (SP142, 22C3, E1L3N, SP263, and NAT105), among which SP142 was the most frequently used antibody. Two studies used multiple antibodies to carry out the IHC tests [39,45] (Figure 11D).

#### 3.4.5. Subgroup Analysis: Regions

We also performed meta-analyses in different regions. Fourteen studies were divided into three subgroups, including North America, Europe, and Asia. The North American group includes three studies, the European group includes three studies, and the Asian group includes eight studies. The pooled expression level of PD-L1 in the North American group (81%) is much higher than the Asian group (65%), but no significant difference is observed (*p* = 0.25, random effect model) (Figure 11E).

## 4. Discussion

As a rare disease, the limited number of TET patients hampered the research of the molecular landscapes of TETs, since the full understanding of the disease is built on the analysis of sufficient samples. In our meta-analysis, 1102 patients are included, among which the largest cohort size of a single study is 264 [22]. Thus, the pooled estimated mutation rates of each gene might be more convincing, considering the enormous number of subjects contained in this study, although high heterogeneity is inevitably observed in some cases. This study has provided a more accurate gene alteration and PD-L1 expression rate from the studies reported so far, thereby deepening the understanding of the molecular alterations in this rare disease.

### 4.1. Gene Alteration Landscape

Gene mutation has been identified as one of the causes of various cancers. The *EGFR* mutation in non-small-cell lung cancer (NSCLC) is one of earliest recognized driver gene mutations in the treatment of cancers. Osimertinib, a third-generation targeted drug aiming at *EGFR* with tyrosine kinase inhibitors (TKIs), has been widely implemented in patients with advanced NSCLC [54]. Although TETs were reported to harbor the lowest tumor mutational burden (TMB) among the common cancers in adults, various gene mutations have been identified in TETs [10]. Unfortunately, targeted therapy with high efficacy has not been established in treating advanced TETs. No drugs specifically targeting actionable gene mutations in TETs have been put into clinical practice [55]. However, some gene mutations were also found to impact the efficacy of other treatments and overall survival. Stockhammer et al. found that patients with *TP53* mutant/*TSG* mutant lung cancers had a worse survival with a lower efficacy in *EGFR*-TKI therapy [56]. Additionally, other genes associated with PD-L1 expression, such as *CYLD* and *BAP1*, contribute to the efficacy of immunotherapy. Patients with *CYLD*-mutated TCs tend to respond more to ICIs than to TCs harboring the *BAP1* mutation [57]. Therefore, gene mutation still plays a significant role in TET treatment, and this meta-analysis endeavored to display the current overall mutational landscape in TETs and to lay the foundations for further research.

Early studies have identified the mutational statuses of the *EGFR*, *KIT*, and *RAS* family genes by direct sequencing and PCR [58,59,60], establishing a critical groundwork for subsequent investigations. Next-generation sequencing and its derived technologies have brought opportunities for a more thorough sequencing of TETs, propelling us toward the overarching goal of achieving a complete elucidation of the genomic landscape of TETs [9].

The *GTF2I* mutation (chr7 c.74146970T>A) was first reported in type A thymomas at a high frequency by Petrini et al., and was found in 82% of type A and 74% of type AB thymomas [19]. Within the framework of the TCGA project, Radovich et al. conducted WES on 117 samples, and detected mutated *GTF2I* in all type A and 70% of type AB thymomas [10]. Thus, the mutation of *GTF2I* stood as a significant hallmark of type A and AB thymomas. In our study, the pooled estimated mutation rate of *GTF2I* in thymomas was 0.4263 (95% CI [0.3590; 0.4936]), with minor heterogeneity, implicating the uniformity of the *GTF2I* mutation among the reported studies.

*TP53*, one of the most frequently mutated tumor suppressor genes among all types of cancer, was also found to be frequently mutated in TETs, especially in TCs. In our study, *TP53* was the second most frequently mutated gene in thymomas. The estimated pooled mutation rate was 0.1101 (95% CI [0.00000; 0.2586]), with a high heterogeneity. A subgroup analysis could not be conducted due to the limited number of studies in several subgroups. However, through the leave-one-out sensitivity analysis, we spotted the study ”Szpechcinski 2022” [35], which influenced the robustness of the pooled results. By comparing the sequencing methods, samples, and statistical analysis of this study, we found that this study reported gene mutations by detecting single-nucleotide variants (SNVs) in genes commonly mutated in solid tumors, while other studies primarily targeted known exons and/or introns in cancer-related genes. This might explain the high mutational rates in this study to some extent. When omitting this study, the mutation rate of *TP53* in thymomas was reduced to 0.0240 (95% CI [0.0082; 0.0399]) (common effect model), with a fairly low heterogeneity (*p* = 0.4029). Meanwhile, TP53 was the most common mutated gene in TCs, with a pooled estimated rate of 0.1797 (95% CI, [0.0732; 0.3203]) and a high heterogeneity. Since most of the studies contained both thymomas and TCs in their cohorts, we were unable to perform a subgroup analysis by their subtypes. However, when separating the TSCC samples from the cohort, we found that the *TP53* mutation rate in TSCCs was 0.1556 [0.0866; 0.2246] (common effect model), with a low heterogeneity (*I*^2^ = 34%, *p* = 0.1923), demonstrating the homogeneity of the *TP53* mutation across the TSCC samples. It can also be concluded that different TC subtypes may have diverse gene mutation patterns. It should also be noted that the *TP53* mutation frequency was significantly higher in Europe and North America than in Asia according to the subgroup analysis by regions.

To sum up, our study systematically reviewed a fairly large number of articles and provided a comprehensive overview of the mutational landscape of frequently mutated genes in TETs. Thus, the reported gene mutational rates in thymomas and TCs provide more convincing proof for the investigations into the targeted therapy of TETs. Moreover, our study also found that sequencing methods and regions may both influence the gene mutational rate, whether in thymomas or in TCs. In clinical practice, when implementing targeted therapy, a proper gene target is a crucial factor impacting the efficiency of targeted drugs and the overall survival of the patients. These two factors should also be taken into consideration in the clinical decision-making process.

### 4.2. PD-L1 Expression Levels

PD-L1 is a member of the B7 family, which is an important mediator of the evasion of tumor immunity [61], and is mainly expressed on tumor cells as well as many hematopoietic cell types [62]. PD-L1 can inhibit immune response by binding to programmed cell death protein 1 (PD-1) on activated T cells. Many studies discovered that a high level of PD-L1 expression was related to a poor prognosis in certain cancers, indicating PD-L1 as a convincing immunotherapy target [62,63]. Although there are not many phase II clinical trials on immunotherapies for thymic epithelial tumors, several phase II clinical trials on the efficacy of immunotherapies in patients with advanced thymic carcinoma still reveal the relationship between PD-L1 and the benefit of immunotherapy. Previous studies have reported that thymic carcinoma patients with a high PD-L1 expression, i.e., at least 50% of the tumor cells being PD-L1-positive, had a higher partial or complete response rate compared to those with a low PD-L1 expression after treatment with pembrolizumab [8,17]. In another phase II study by He et al., a higher PD-L1 expression level in advanced TC patients was correlated with the better efficacy of pembrolizumab [57]. These studies have shown that the expression level of PD-L1 in tumor cells is related to the efficacy of the immunotherapy. However, due to the rarity of thymic neoplasms, the data on PD-L1 expression levels are still limited. Previous studies on TETs reported on the varieties of PD-L1 expression levels. Thus, we conducted this meta-analysis study to obtain a relatively reliable expression level of PD-L1 in thymic epithelial tumors.

The overall PD-L1 expression level in our analysis was 71% (95% CI [0.59; 0.81], *p* < 0.01, random effect model), with a high heterogeneity. Thus, the subgroup analysis was preformed according to the following aspects.

Our meta-analysis indicates that the PD-L1-positive rate is higher in the thymoma group than the TC group in the low-threshold group (0.83 vs. 0.71), despite the insignificant difference (*p* = 0.19). In the high-threshold group, however, the two rates come closer together (0.41 vs. 0.42, *p* = 0.87). Many studies have shown that different subtypes of thymomas have diverse PD-L1 expression statuses. Generally, B2/B3 thymomas exhibit higher PD-L1 expression levels than A/AB/B1 thymomas [42,46]. Unfortunately, we were unable to ascertain the PD-L1 expression levels in different subtypes of thymomas, which is a limitation of this study. However, our study shows that thymomas might present a higher PD-L1 expression level than TCs.

PD-L1 has become an outstanding prognostic factor and immunotherapy target, with various antibodies developed by companies to detect the PD-L1 expression in IHC tests. From a clinical point of view, differences in the PD-L1 expression level caused by the antibody selection will affect the judgment of clinicians on the effect of immunotherapy. From the perspective of clinical trial design, the use of different antibodies in the immunohistochemical staining of tissues may be a confounding factor in the study. Therefore, it is meaningful to determine whether different antibodies will affect the expression level of PD-L1 in the same tissue. Though studies utilizing SP142 as the primary antibody present a lower PD-L1-positive ratio compared to other antibodies in the low-threshold group (SP142 vs. 22C3 and 69% vs. 77%), the result is the opposite in the high-threshold group, whereas studies utilizing SP142 presented the highest PD-L1-positive ratio of 53% compared with the other three groups (E1L3N: 43%, SP263: 39%, and 22C3: 41%). The difference is significant between groups, indicating that the choice of antibody may exert an influence on the results.

Other factors such as regions, the Masaoka stage, and the treatment may be related to the PD-L1 expression level and contribute to the heterogeneity, which cannot be ignored.

A subgroup analysis was also performed based on the region, but there was no group difference. It is noteworthy that the Asian group showed a high heterogeneity, indicating that the heterogeneity brought by the regions cannot be neglected.

The effect of the Masaoka stage on the PD-L1 expression was supported in several studies. Bedekovics et al. reported that the PD-L1-positive rate in Masaoka–Koga stage I/II thymic epithelial tumors was higher than Masaoka–Koga stage III/IV thymic epithelial tumors (51.9% vs. 0%) [49]. Hakiri et al. reported the opposite result, where the PD-L1-positive rate was higher in Masaoka–Koga stage III/IV than in Masaoka–Koga stage I/II thymic epithelial tumors (63.6% vs. 21.4%) [40]. Although the results of the two studies are completely opposite, they are still sufficient to show that the Masaoka–Koga stage has an effect on the expression of PD-L1 in thymic epithelial tumors. A larger sample size and more rigorously designed cohort study may be needed to clarify its impact.

Katsuya et al. found that surgery specimens showed a higher level of PD-L1 expression both in the tumor proportion and tumor staining intensity [64]. They also reported an increase in the PD-L1 expression level after chemotherapy compared with before the treatment (100% vs. 67%).

In conclusion, this meta-analysis evaluated the PD-L1 expression levels in TETs. However, different cut-off values, subtypes, antibodies utilized in the IHC tests, regions, the Masaoka stage, and treatment can exhibit a great influence on the overall results. Further investigations are necessary to determine how these factors can be mitigated to enhance the accuracy of the PD-L1 expression levels and improve the comparability across all TET patients. Additionally, physicians should not overlook the potential impact these factors may have on the efficacy of ICI-based immunotherapy in clinical practice.

### 4.3. Limitations

We performed a meta-analysis reviewing previously reported gene mutational rates and PD-L1 expression levels. A large number of studies and samples were reviewed, which improved the reliability of our results. Possible sources of heterogeneity were also explored in the study through a subgroup analysis and sensitivity analysis. However, high heterogeneity could not be avoided in several analyses, which were acceptable since it was not binary outcome. However, we have to admit the restriction of these pooled results. Additionally, in the screening process, we only included articles written in English, which can hamper the inclusion of possible results and impact the accuracy of the pooled results. Of note, in our subgroup analysis, the subgroups for the sequencing methods only included the “WES” group and the “targeted NGS” group. A more detailed classification might improve the reliability of the final results.

## 5. Conclusions

In brief, this study provides an overview of the gene alteration landscape and PD-L1 expression levels in TETs, with a uniform *GTF2I* mutational rate in thymomas and *TP53* mutational rate in TSCCs. Confounding factors, such as cut-off values, subtypes, antibodies, regions, the Masaoka stage, and treatment, can all contribute to the heterogeneity in the PD-L1 expression levels. Certain limitations do exist, potentially impacting the accuracy of our findings, necessitating further in-depth studies to pave the way for more effective strategies in treating TETs.

## Figures and Tables

**Figure 1 cancers-16-02966-f001:**
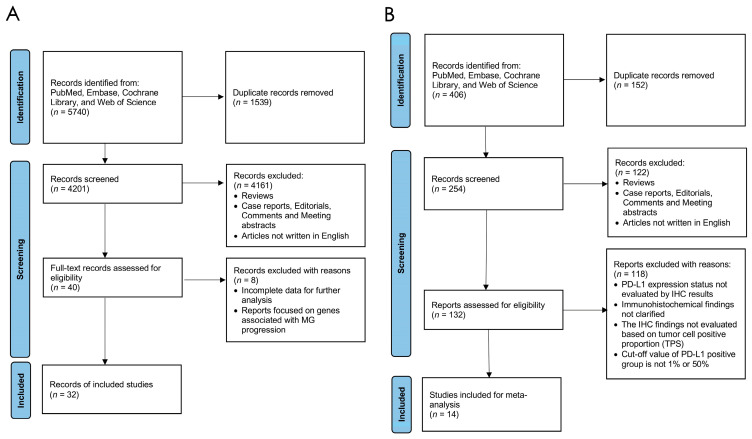
PRISMA flow chart depicting the literature searching process in the gene alteration cohort (**A**) and the PD-L1 expression level cohort (**B**).

**Figure 2 cancers-16-02966-f002:**
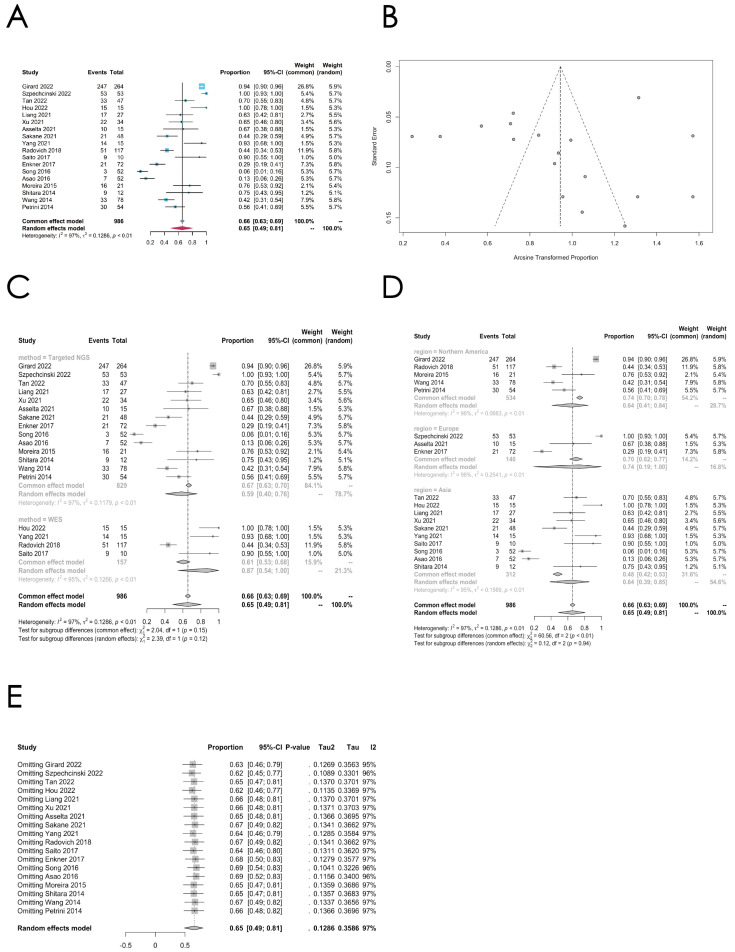
Meta-analysis, subgroup analysis, and sensitivity analysis of the overall mutational rate in TETs. (**A**) Forest plot of the overall mutation rate in TETs. (**B**) Funnel plot of the overall mutation rate in TETs. (**C**) Subgroup analysis by sequencing methods of the overall mutation rate in TETs. (**D**) Subgroup analysis by regions of the overall mutation rate in TETs. (**E**) Sensitivity analysis of the overall mutation rate in TETs.

**Figure 3 cancers-16-02966-f003:**
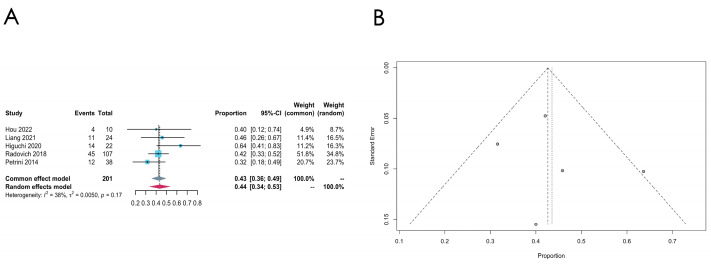
(**A**,**B**) Meta-analysis of the *GTF2I* mutation in thymomas. (**A**) Forest plot of the *GTF2I* mutation in thymomas. (**B**) Funnel plot of *GTF2I* in thymomas.

**Figure 4 cancers-16-02966-f004:**
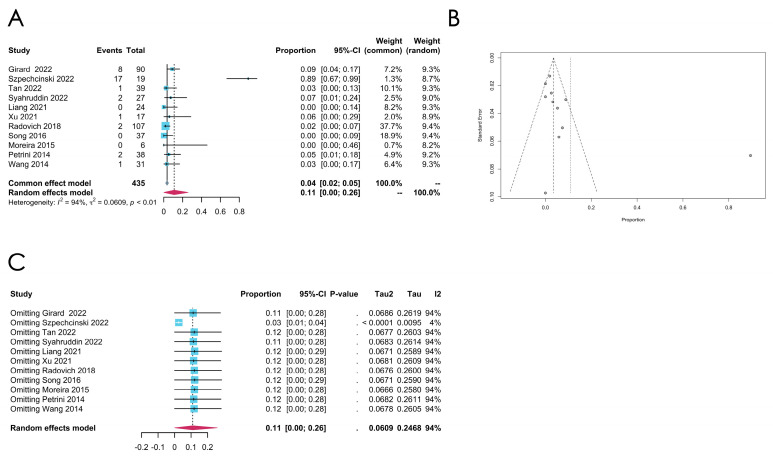
(**A**–**C**) Meta-analysis of the *TP53* mutation in thymomas. (**A**) Forest plot of the *TP53* mutation in thymomas. (**B**) Funnel plot of the *TP53* in thymomas. (**C**) Sensitivity analysis of the *TP53* mutation in thymomas.

**Figure 5 cancers-16-02966-f005:**
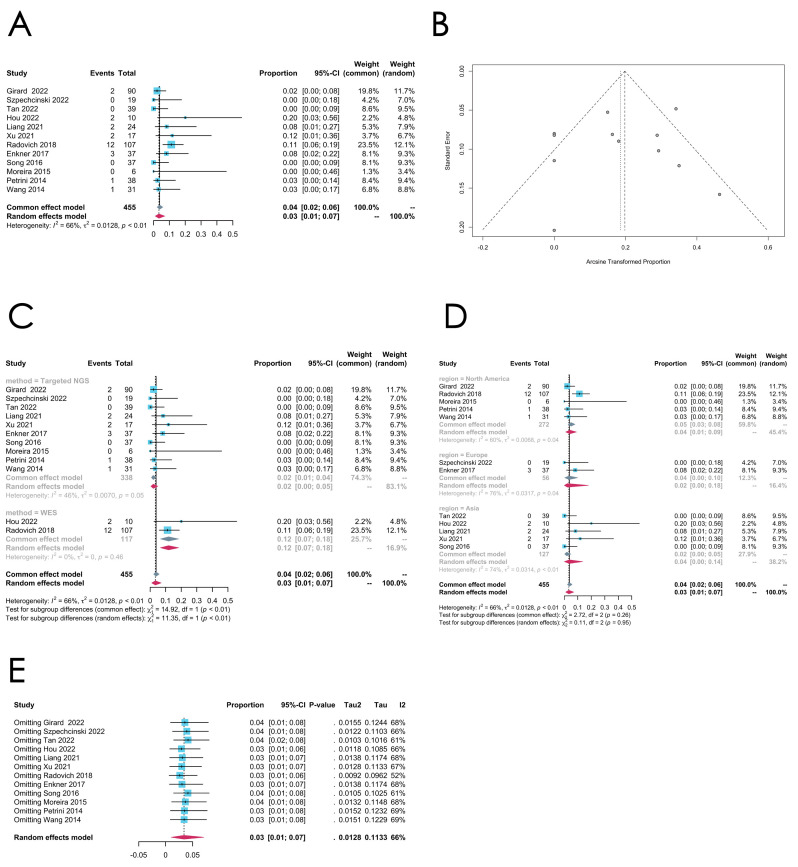
(**A**–**E**) Meta-analysis, subgroup analysis, and sensitivity analysis of the *RAS* family gene mutation in thymomas. (**A**) Forest plot of the *RAS* family gene mutation in thymomas. (**B**) Funnel plot of the *RAS* family gene mutation in thymomas. (**C**) Subgroup analysis by sequencing methods of the *RAS* family gene mutation in thymomas. (**D**) Subgroup analysis by regions of the *RAS* family gene mutation in thymomas. (**E**) Sensitivity analysis of the *RAS* family gene mutation in thymomas.

**Figure 6 cancers-16-02966-f006:**
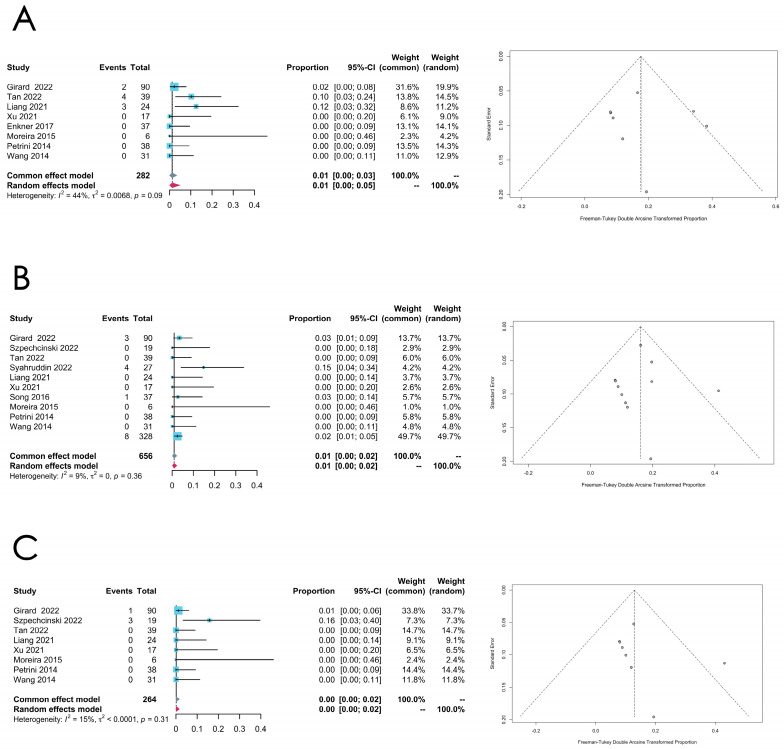
(**A**–**C**) Meta-analysis of the *ATM*, *EGFR*, and *KIT* mutations in thymomas. (**A**) Forest plot and funnel plot of the *ATM* mutation in thymomas. (**B**) Forest plot and funnel plot of the *EGFR* mutation in thymomas. (**C**) Forest plot and funnel plot of the *KIT* mutation in thymomas.

**Figure 7 cancers-16-02966-f007:**
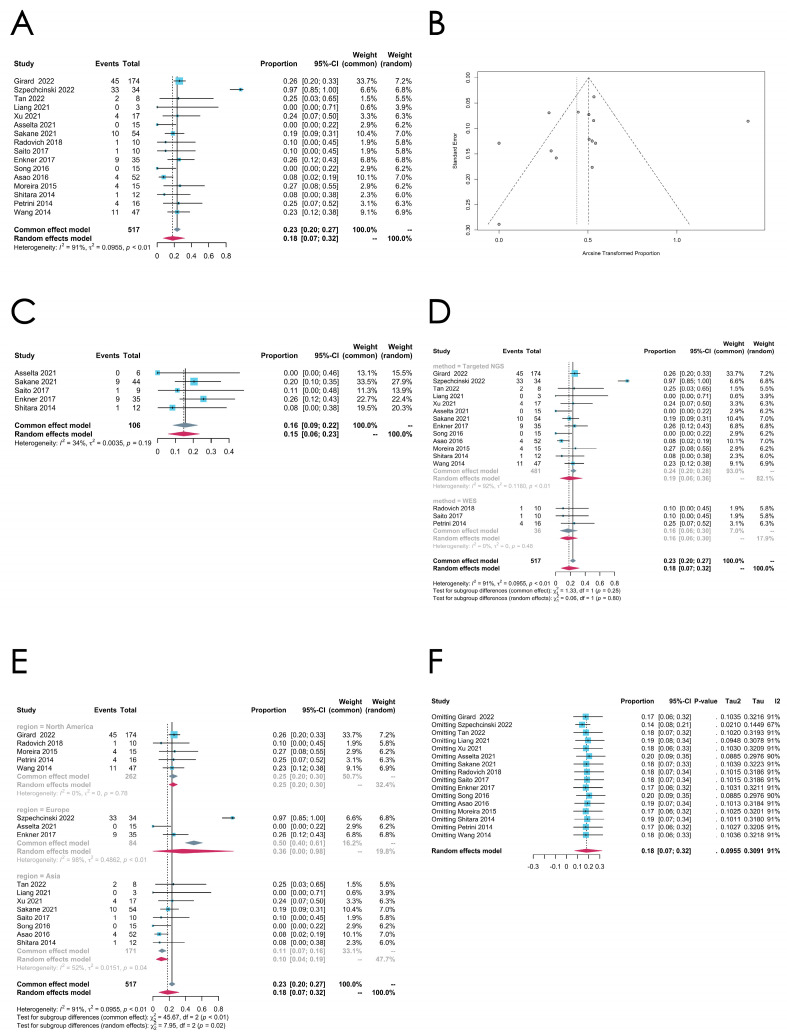
(**A**–**F**) Meta-analysis, subgroup analysis, and sensitivity analysis of the *TP53* mutation in thymic carcinomas (TCs). (**A**) Forest plot of the *TP53* mutation in TCs. (**B**) Funnel plot of the *TP53* mutation in TCs. (**C**) Forest plot of the *TP53* mutation in thymic squamous cell carcinomas (TSCCs). (**D**) Subgroup analysis by sequencing methods of the *TP53* mutation in TCs. (**E**) Subgroup analysis by regions of the *TP53* mutation in TCs. (**F**) Sensitivity analysis of the *TP53* mutation in TCs.

**Figure 8 cancers-16-02966-f008:**
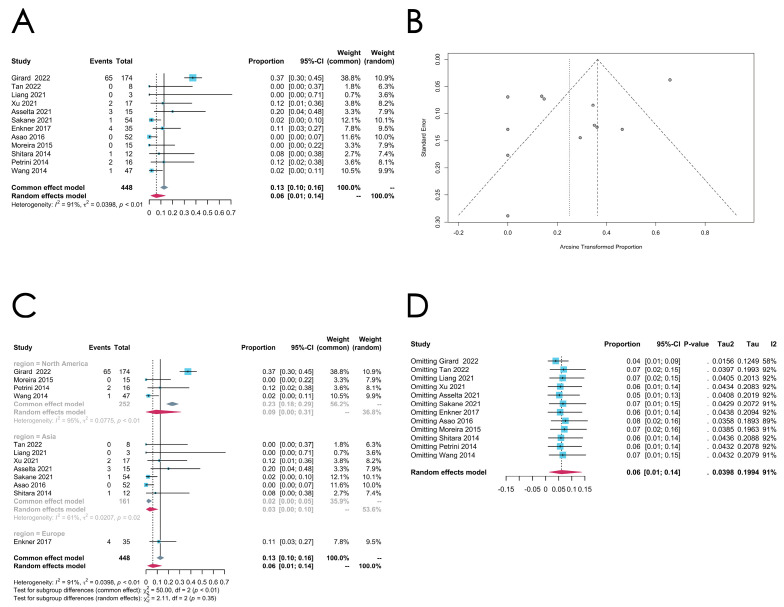
(**A**–**D**) Meta-analysis, subgroup analysis, and sensitivity analysis of the *CDKN2A* mutation in thymic carcinomas (TCs). (**A**) Forest plot of the *CDKN2A* mutation in TCs. (**B**) Funnel plot of the *CDKN2A* mutation in TCs. (**C**) Subgroup analysis by regions of the *CDKN2A* mutation in TCs. (**D**) Sensitivity analysis of the *CDKN2A* mutation in TCs.

**Figure 9 cancers-16-02966-f009:**
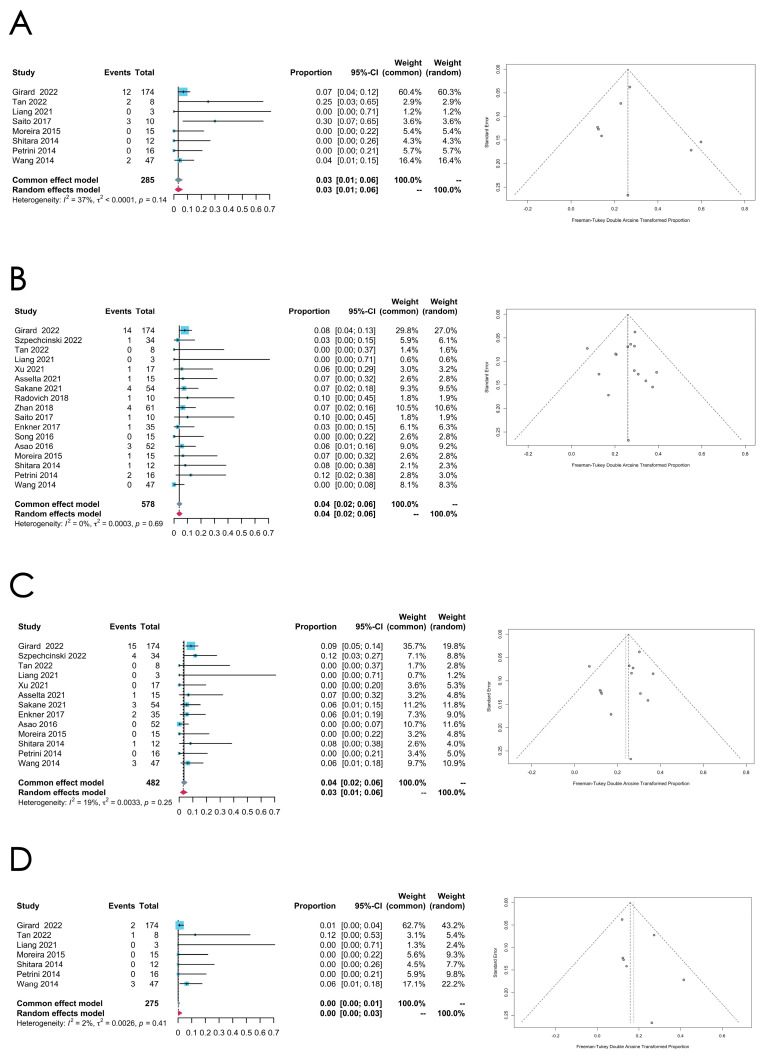
(**A**–**D**) Meta-analysis of the *TET2*, *RAS* family gene, *KIT*, and *DNMT3A* mutations in thymic carcinomas (TCs). (**A**) Forest plot and funnel plot of the *TET2* mutation in TCs. (**B**) Forest plot and funnel plot of the *RAS* family gene mutation in TCs. (**C**) Forest plot and funnel plot of the *KIT* mutation in TCs. (**D**) Forest plot and funnel plot of the *DNMT3A* mutation in TCs.

**Figure 10 cancers-16-02966-f010:**
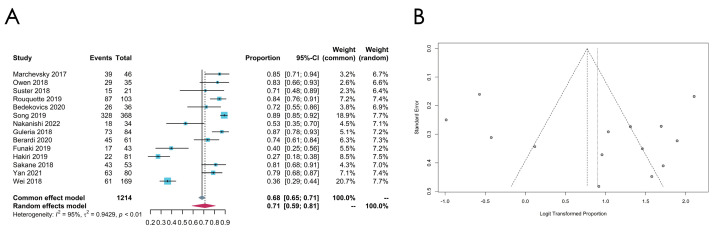
(**A**,**B**) Meta-analysis of the overall PD-L1 expression levels in TETs. (**A**) Forest plot of the overall PD-L1 expression levels in TETs. (**B**) Funnel plot of the overall PD-L1 expression levels in TETs.

**Figure 11 cancers-16-02966-f011:**
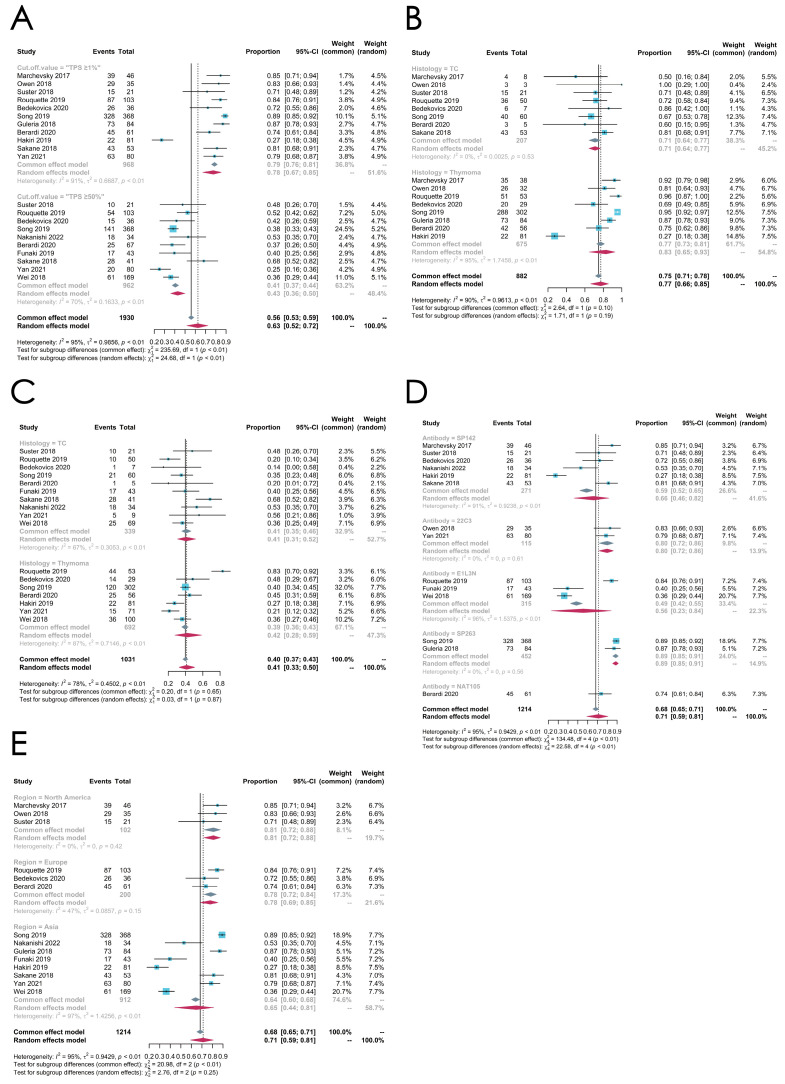
(**A**–**E**) Subgroup analyses of PD-L1 expression levels in TETs by different standards. (**A**) Subgroup analysis by cut-off values. (**B**) Subgroup analysis by histologic subtypes in the low-threshold group. (**C**) Subgroup analysis by histologic subtypes in the high-threshold group. (**D**) Subgroup analysis by antibodies (SP142, 22C3, E1L3N, SP263, and NAT105). (**E**) Subgroup analysis by regions (North America, Europe, and Asia).

**Table 1 cancers-16-02966-t001:** An overview of the features of the selected studies in the gene alteration cohort. TC: thymic carcinoma; NGS: next-generation sequencing; WES: whole-exon sequencing; PCR: polymerase chain reaction.

Study	Region	Cohort Size	WHO Histologic Classification	Number of Subjects	Sequencing Method	Gene Panel	Genes Detected
Girard et al., 2022 [22]	US	264	Thymoma TC	174 90	Targeted NGS	318 cancer-related genes plus 37 introns from 28 genes frequently rearranged in cancer	*CDKN2A*, *TP53*, *KIT*, *RAS*, etc.
Szpechcinski et al., 2022 [35]	Poland	53	Thymoma TC	19 34	Targeted NGS	15 genes implicated in common solid tumors	*ERBB2*, *KIT*, *KRAS*, *FOXL2*, etc.
Tan et al., 2022 [25]	China	47	Thymoma TC	39 8	Targeted NGS	315 cancer-related genes	*TP53*, *TET2*, *ATM*, *DNMT3A*, etc.
Syahruddin et al., 2022 [37]	Indonesia	27	Thymoma	27	PCR	*TP53* and *EGFR*	*TP53*, *EGFR*
Hou et al., 2022 [20]	China	15	Thymoma TC	10 5	WES	-	*GTF2I*, *HRAS*, *NBPF14*, etc.
Liang et al., 2021 [38]	China	27	Thymoma TC	24 3	Targeted NGS	a 476-gene panel	*ATM*, *NF1*, *GTF2I*, *NRAS*, etc.
Xu et al., 2021 [23]	China	34	Thymoma TC	17 17	Targeted NGS	a panel of 56 cancer-related genes	*TP53*, *MTOR*, *CDKN2A*, etc.
Asselta et al., 2021 [30]	Italy	15	TC	15	Targeted NGS	50 onco- and tumor-suppressor genes	*CDKN2A*, *RAS*, *FGFR3*, etc.
Sakane et al., 2021 [31]	Japan	54	TC	54	Targeted NGS	50 of the most commonly reported oncogenes and tumor suppressor genes	*PDGFRA*, *KIT*, *PIK3CA*, *RAS*, etc.
Yang et al., 2021 [21]	China	15	Thymoma TC	6 9	WES	-	*ZNF429*, *ZNF208*, *ERBB4*, etc.
Higuchi et al., 2020 [33]	Japan	22	Thymoma	22	Targeted sequencing	*GTF2I*	*GTF2I*
Radovich et al., 2018 [10]	US	117	Thymoma TC	107 10	WES	-	*GTF2I*, *RAS*, *TP53*
Zhan et al., 2018 [29]	China	61	TC	61	PCR and direct sequencing	*EGFR* pathway genes	*EGFR*, *PI3KCA*, *BRAF*, etc.
Saito et al., 2017 [36]	Japan	10	TC	10	WES	-	*TP53*, *RAS*, *CYLD*, *SETD2*, etc.
Enkner et al., 2017 [28]	Austria	72	Thymoma TC	37 35	Targeted NGS	50 onco- and tumor-suppressor genes	*RAS*, *SMARCB1*, *CDKN2A*, *FGFR3*, etc.
Song et al., 2016 [34]	China	52	Thymoma TC	37 15	Targeted NGS	a panel of 22 genes	*EGFR*, *PIK3CA*, etc.
Asao et al., 2016 [27]	Japan	52	TC	52	Targeted NGS	50 cancer-related genes	*TP53*, *FBXW7*, etc.
Moreira et al., 2015 [26]	US	21	Thymoma TC	6 15	Targeted NGS	275 commonly implicated oncogenes, tumor suppressor genes, and members of pathways considered actionable by targeted therapies	*BCOR*, *MLL3*, *RAS*, *KDM6A*, etc.
Shitara et al., 2014 [24]	Japan	12	TC	12	Targeted NGS	409 tumor suppressor genes and oncogenes	*NF1*, *KIT*, *CDKN2A*, *TET1*, *TP53*, etc.
Petrini et al., 2014 [19]	US	54	Thymoma TC	38 16	WES and Targeted NGS	197-gene assay	*GTF2I*, *TP53*, *CDKN2A*, etc.
Wang et al., 2014 [32]	US	78	Thymoma TC	31 47	Targeted NGS	197 cancer-related genes	*DCC*, *RAS*, *SETD2*, *BAP1*, etc.

**Table 2 cancers-16-02966-t002:** An overview of the features of the selected studies in the PD-L1 expression level cohort. TC: thymic carcinoma; IHC: immunohistochemistry; NA: not available.

Study	Region	Cohort Size	WHO Histologic Classification	Cut-Off Value for Positive	Sample Source	Treatment	Antibody for IHC
Marchevsky et al., 2017 [47]	US	46	A, AB, B1-3, TC	1%	Surgical resection	NA	SP142
Owen et al., 2018 [44]	US	35	A, AB, B1-3, TC	1%	Surgical resection	NA	22C3
Suster et al., 2018 [43]	US	21	TC	1% and 50%	Surgical resection	NA	SP142
Rouquette et al., 2019 [45]	France	103	B3, TC	1% and 50%	Surgical resection and biopsy	NA	E1L3N, 22C3, SP142, SP263
Bedekovics et al., 2020 [49]	Hungary	36	A, AB, B1-3, TC	1% and 50%	Surgical resection	preoperative radio-based/cisplatin-based chemotherapy	SP142
Song et al., 2019 [41]	South Korea	368	A, AB, B1-3, TC	1% and 50%	Surgical resection	neoadjuvant and/or postoperative adjuvant radiation or chemotherapy	SP263
Nakanishi et al., 2022 [50]	Japan	34	TC	50%	Surgical resection	NA	SP142
Guleria et al., 2018 [46]	India	84	A, AB, B1-3	1%	Surgical resection and biopsy	NA	SP263
Berardi et al., 2020 [48]	Italy	61	A, AB, B1-3, TC	1% and 50%	Surgical resection and biopsy	NA	NAT105
Funaki et al., 2019 [52]	Japan	43	TC	50%	Surgical resection	preoperative radio-based/cisplatin-based chemotherapy	E1L3N
Hakiri et al., 2019 [40]	Japan	81	A, AB, B1-3	1%	Surgical resection	NA	SP142
Sakane et al., 2018 [39]	Japan	53	TC	1% and 50%	Surgical resection and biopsy	chemotherapy/radiotherapy	SP142, SP263, 22C3, 28-8
Yan et al., 2021 [51]	China	80	A, AB, B1-3, TC	1% and 50%	Surgical resection	NA	22C3
Wei et al., 2018 [42]	China	169	A, AB, B1-3, TC	50%	Surgical resection and biopsy	preoperative radio-based/cisplatin-based chemotherapy	E1L3N

**Table 3 cancers-16-02966-t003:** The summary of the results of the meta-analysis on the prevalence of gene mutations in thymomas and thymic carcinomas.

Subtype	Gene	Number of Included Studies	Pooled Estimated Rate (95% Confidence Interval)	*I* ^2^	*p*-Value	Model	Transformation Method	Egger’s Test (*p*-Value)
Thymoma	*GTF2I*	5	0.4263 [0.3590;0.4936]	38%	0.1657	Common effect model	Untransformed	0.6476
	*TP53*	11	0.1101 [0.0000; 0.2586]	94%	<0.0001	Random effect model	Untransformed	0.1139
	*RAS*	12	0.0341 [0.0104; 0.0710]	66%	0.0007	Random effect model	Arcsine	0.6011
	*ATM*	8	0.0121 [0.0003; 0.0342]	44%	0.0855	Common effect model	Freeman–Tukey double arcsine	0.8333
	*EGFR*	10	0.0093 [0.0014; 0.0218]	9%	0.3584	Common effect model	Freeman–Tukey double arcsine	0.8802
	*KIT*	8	0.0011 [0.0000; 0.0157]	15%	0.3111	Common effect model	Freeman–Tukey double arcsine	0.4357
Thymic Carcinoma	*TP53*	16	0.1797 [0.0732; 0.3203]	91%	<0.0001	Random effect model	Arcsine	0.4031
	*CDKN2A*	12	0.0608 [0.0139; 0.1378]	91%	<0.0001	Random effect model	Arcsine	0.0654
	*TET2*	8	0.0318 [0.0087; 0.0639]	37%	0.1363	Common effect model	Freeman–Tukey double arcsine	0.7948
	*RAS*	17	0.0389 [0.0201; 0.0616]	0%	0.6887	Common effect model	Freeman–Tukey double arcsine	0.9315
	*KIT*	13	0.0352 [0.0164; 0.0586]	19%	0.2496	Common effect model	Freeman–Tukey double arcsine	0.3152
	*DNMT3A*	7	0.0001 [0.0000; 0.0127]	2%	0.4103	Common effect model	Freeman–Tukey double arcsine	0.2678

## Data Availability

The raw data supporting the conclusions of this article will be made available by the authors on request.

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
