# Peer review of "Depiction of the Genetic Alterations and Molecular Landscapes of Thymic Epithelial Tumors: A Systematic Review and Meta-Analysis"

_cancers, 2024, doi:10.3390/cancers16172966_

Round 1

Reviewer 1 Report

Comments and Suggestions for Authors

The manuscript by Li et al. entitled "Depiction of the Genetic Alterations and Molecular Landscapes 2 of Thymic Epithelial Tumors: a Systematic Review and Meta Analysis" summarizes the genetic mutational landscape and PD-L1 expression in thymic epithelial tumors. The authors have selected two independent subjects for a systematic review: analysis of the major genetic alterations in TETs and PD-L1 expression. It is not clear how these two issues are related and why they should be highlighted in a single publication. The aim of the review is not clearly stated. No correlation between PD-L1 expression and other parameters (e.g., survival or treatment response) was assessed, which seems to be of huge interest in the context of the effectiveness of ICI therapy. Was PD-L1 expression assessed in malignant cells or TME?

Issues to be addressed:

-         Abstract: gene names should be italicized. Check that throughout the entire manuscript  

-         Abstract: TSCC abbreviation is not provided in full

-         Title: a focus on PD-L1 expression is not reflected in the title

-         Materials and Methods: manufacturer, city and county for R software

-         Results: Quality of Figure 1 should be improved. Inscriptions are hardly readable.

-         Tables 1-3. Provide the list of abbreviations used for each table

-         Tables 1-2. Were all the papers included in the study single-authored? It is suggested to indicate a study as, e.g., Girard et al., 2022 in Tables

-         Line 50: Remove “and” and add a dot after etc.

Reviewer 2 Report

Comments and Suggestions for Authors

This is an interesting meta-analysis investigating the dynamic genomic alternations and the expression levels of PD-L1 in TET patients of different subtypes.

The meta-analysis is well conducted and can have some relevance in the current scientific debate. Some comments:

- the comment on immunotherapy in the Introduction is very cursory. can the Authors expand, and further clarify the rationale for the present analysis?

- lines 198-end of the paper: the same statements are reapeated. can the Authors revise and - importantly - suggest clear lines of future research?

- quality of figures if very poor

Comments on the Quality of English Language

moderate editing is required.

Reviewer 3 Report

Comments and Suggestions for Authors

The review article titled "Depiction of the Genetic Alterations and Molecular Landscapes of Thymic Epithelial Tumors: a Systematic Review and Meta-Analysis" presents a thorough and methodologically rigorous meta-analysis of genetic alterations and PD-L1 expression levels in thymic epithelial tumors (TETs). The comprehensive literature review, adherence to PRISMA guidelines, and robust statistical analyses using R software lend credibility to the findings. The results are detailed and well-illustrated with forest plots and subgroup analyses, effectively addressing high heterogeneity and potential confounding factors. The discussion contextualizes the findings within the broader field, highlighting clinical implications and identifying key mutated genes such as GTF2I and TP53. Acknowledging limitations, the manuscript suggests future large-scale studies to validate results and standardize methodologies.

To enhance the paper, it would be beneficial to include a more detailed exploration of the potential clinical applications of these findings and consider integrating additional recent studies to further support the conclusions. Additionally, clarifying the impact of different antibodies on PD-L1 expression results could improve the robustness of the conclusions.

Reviewer 4 Report

Comments and Suggestions for Authors

Wang et al. conducted a meta-analysis of 21 studies on genomic alterations and 14 studies on PD-L1 expression levels in thymic epithelial tumors (TETs). They found the top mutated genes in thymomas and thymic carcinomas were GTF2I, TP53, and RAS, with pooled PD-L1 expression levels at 71%. This review highlights the genetic landscape and potential confounding factors in TETs, aiding further molecular investigations. I have few queries in this paper, can authors address them?

1) The inclusion criteria for selecting studies should be explicitly detailed. Specifically, clarify any thresholds for gene mutation rates or PD-L1 expression levels used to include or exclude studies from the meta-analysis.

2) The manuscript mentions potential confounding factors contributing to heterogeneity. Provide a more in-depth analysis of these factors and their possible impacts on the study results.

3) Elaborate on the data extraction process. For instance, explain how disagreements between reviewers were resolved and the criteria used to judge the quality of the included studies.

4) My biggest concern is text readability in the figures. Please separate the multicomponent figures and present them individually to improve readability. Ex: Fig. 6, 4, 3, etc. 

5) While subgroup analyses were conducted, the rationale for the chosen subgroups (e.g., sequencing methods, cut-off values for PD-L1 expression) should be more thoroughly justified in the methods section.

6) Include additional forest plots and sensitivity analyses for the top mutated genes in both thymomas and thymic carcinomas. This will help illustrate the robustness of your findings and the degree of heterogeneity.

7) The discussion should more explicitly address the clinical implications of your findings. For instance, how might the pooled mutation rates of specific genes influence the choice of targeted therapies for TETs?

8) The high heterogeneity observed in several meta-analyses needs a more detailed examination. Explain potential sources of this heterogeneity and how they were addressed.

Comments on the Quality of English Language

Minor editing of English needed.

Reviewer 5 Report

Comments and Suggestions for Authors

1.    The background provided on thymic epithelial tumors is superficial and does not adequately explain the clinical relevance or need for this study.

2.    The section on genetic characteristics of TETs lacks in depth discussion of the implications of these genetic alterations.

3.    The manuscript mentions high heterogeneity in mutation rates but does not adequately discuss potential sources or implications of this heterogeneity.

4.    The presentation of overall mutation rates is confusing and lacks a clear explanation of the significance and implications of these rates for the field.

5.     The analysis of specific gene mutations (e.g., GTF2I, TP53) does not adequately explore the biological or clinical relevance of these mutations.

6.    The section on PD-L1 expression is not detailed enough and fails to provide a comprehensive analysis of its prognostic value and implications for treatment.

7.    The manuscript notes significant regional variations in study populations but does not discuss how these might impact the generalizability of the findings.

Comments on the Quality of English Language

Minor editing of English language required

Round 2

Reviewer 1 Report

Comments and Suggestions for Authors

-         The authors have addressed the comments  

Comments on the Quality of English Language

In general, the language is fine

Reviewer 4 Report

Comments and Suggestions for Authors

The authors have addressed my comments.

Reviewer 5 Report

Comments and Suggestions for Authors

Accepted for publication

Comments on the Quality of English Language

Minor editing of English language required.